# Site-specific fabrication of a melanin-like pigment through spatially confined progressive assembly on an initiator-loaded template

Haejin Jeong[1], Jisoo Lee[1], Seunghwi Kim[1], Haeram Moon[1] & Seonki Hong [1] ✉

Melanin-like nanomaterials have emerged in surface biofunctionalization in a material-independent manner due to their versatile adhesion arising from their catechol-rich structures. However, the unique adhesive properties of these materials ironically raise difficulties in their site-specific fabrication. Here, we report a method for site-specific fabrication and patterning of melanin-like pigments, using progressive assembly on an initiator-loaded template (PAINT), different from conventional lithographical methods. In this method, the local progressive assembly could be naturally induced on the given surface pretreated with initiators mediating oxidation of the catecholic precursor, as the intermediates generated from the precursors during the progressive assembly possess sufficient intrinsic underwater adhesion for localization without diffusion into solution. The pigment fabricated by PAINT showed efficient NIR-to-heat conversion properties, which can be useful in biomedical applications such as the disinfection of medical devices and cancer therapies.

Melanin-like nanomaterials such as polydopamine and poly-norepinephrine have emerged in surface functionalization and interfacial engineering due to their unique underwater adhesive properties[1–3]. Contrary to conventional specificity-based surface chemistries, such as self-assembled monolayers (SAMs), silane coating, and layer-by-layer (LbL) deposition, these materials possess a versatile affinity to a variety of substrates arising from catechol-rich characteristics similar to mussel adhesive proteins[4,5]. Owing to this versatility, these materials provide a universal toolkit for engineering solid/water interfaces of various substrates, particularly for (1) controlling the surface wettability, roughness, and hardness[6–9], (2) immobilizing secondary functionalities such as therapeutics, antibodies, and catalytic nanoparticles[10,11], and (3) solving the biointerfacial problems between cells and foreign materials[12–14].

The synthesis of these catechol-rich materials is called a progressive assembly (Fig. 1a)[15]. This assembly process starts with the oxidation of monomeric precursors (phenolic and/or catecholic

derivatives), which further induces intramolecular and/or inter-molecular covalent reactions that form a heterogeneous set of oligomers. During synthesis, these oligomers eventually participate in noncovalent interactions, forming an amorphous nanoscale carbon material. Due to the short lifetime prior to participating in noncovalent interactions, the molecular weight distribution and chemical structure of these oligomeric intermediates have not been fully understood. Therefore, it is highly challenging to tune the coating properties by controlling the as-prepared oligomeric intermediates or nanoscale assemblies, and instead, changes in the starting conditions (i.e., chemical structure of the starting monomers and/or their oxidizing conditions (pH, oxidant type, stirring speed, temperature, etc.) and the existence of additives) are the only available options in current approaches[1–3]. The in situ generation of an uncontrollable set of heterogeneous building blocks for noncovalent assembly distinguishes the progressive assembly process from other conventional self-assembly processes, such as lipid bilayers, microtubules, DNA

[1]Department of Physics and Chemistry, DGIST, Daegu 42988, Republic of Korea. ✉e-mail: seonkihong@dgist.ac.kr

**a**  **Progressive assembly forming polydopamine**

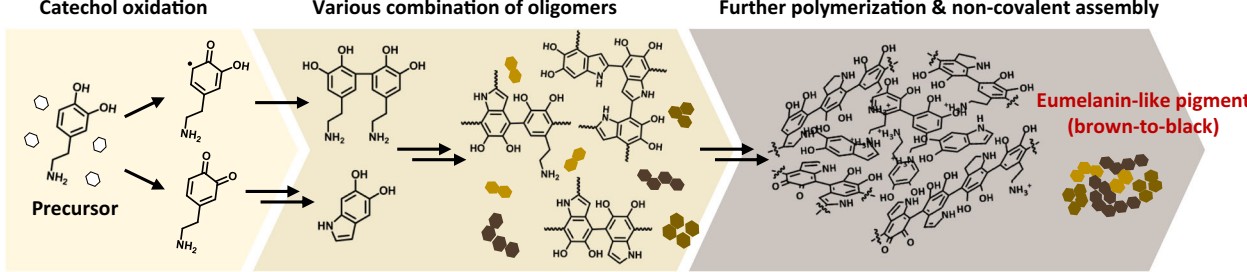

**b**  **Melanogenesis in nature**

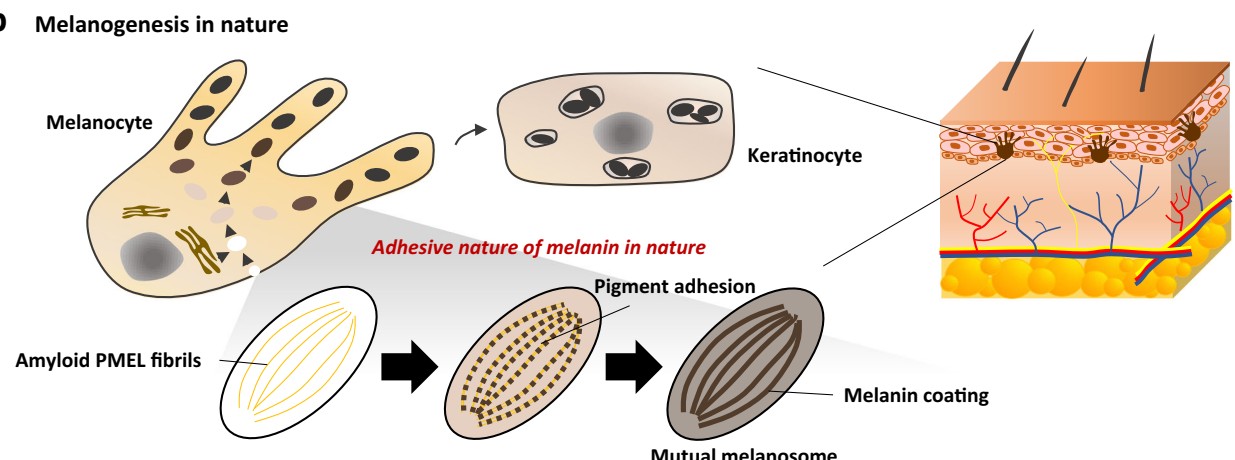

**c**

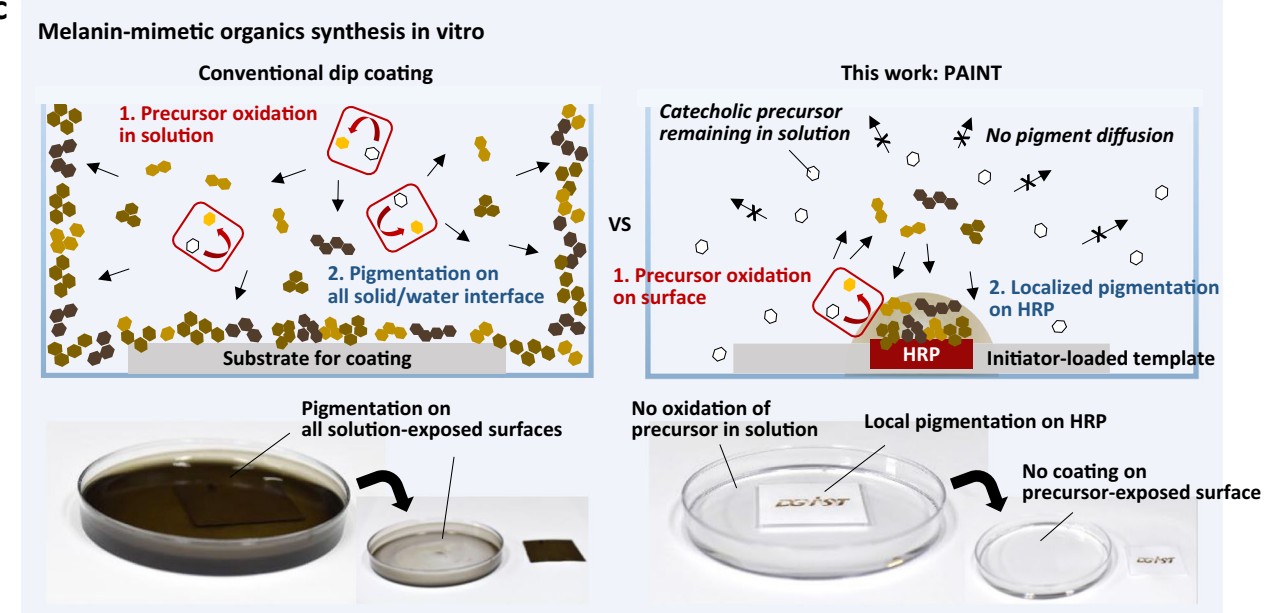

**Fig. 1 | Description of melanogenesis in nature and the in vitro synthesis of melanin-mimetic pigments. a** A plausible synthetic pathway of polydopamine, a well-known synthetic analog of melanin, previously termed progressive assembly[15]. **b** A schematic illustration of natural melanogenesis indicating the adhesive nature of melanin leading to its growth attached to fibrils in melanosomes. **c** Comparison of the in vitro synthetic procedure of melanin-like organic coating from a conventional dip coating method to the surface-initiated in situ pigmentation method called PAINT (Progressive Assembly on an Initiator-loaded Template) reported in this study. On-site pigmentation in PAINT is guided by a surface-immobilized initiator, horseradish peroxidase (HRP), mediating catechol oxidation.

origami, and SAMs, in which presynthesized monotonic building blocks are used[15]. This leads to natural melanin and melanin-like materials possessing a disordered hierarchy, such as chemical structural disorder at the monomer-to-oligomer, molecular-to-electronic, and supramolecular levels, which notably impacts their unique biochemical properties[16].

By revisiting melanogenesis in nature, we were able to draw attention to the amyloid fibrils that act as templates, as shown in Fig. 1b. Melanin synthesis in nature is not a simple solution-phase reaction, which is different from other polymerization reactions. Someone may consider spherical particulates as forms of natural melanins, but in fact, these are mostly melanin-containing

melanosomes, not pure melanin pigments. Instead, true melanin pigments are found to be grown attached to these amyloid fibrils in melanosomes. More interestingly, the localized synthesis on templates was mediated by the catalytic activity of fibrils that accelerates melanin synthesis and is believed to prevent the diffusion of reactive intermediates out of the melanosome by binding and sequestering[17,18]. Similarly, melanin-like materials can be grown as a nano-thin coating attached to all kinds of surfaces exposed to the reaction solution, including the coating solution containers[4,19,20]. Therefore, a variety of different kinds of substrates can be coated at once in a single coating batch, which is the biggest feature of the coating based on melanin-like materials. However, this, on the other hand, has raised the need for site-specific fabrication/patterning techniques in the desired area. To achieve this, the conventional lithographic technique has been employed mostly to spatially confine the area of contact with the coating solution for site-specific fabrication. For example, Kang et al. showed the fabrication of a hydrophilic polydopamine pattern on a superhydrophobic surface by using a polydimethylsiloxane (PDMS) microchannel for water droplet guiding[21]. Shi et al. achieved a gradient polydopamine pattern for controlling cellular behavior by gradually diluting the dopamine solution in a microfluidic channel[22]. However, these techniques are often limited by high instrumental costs and labor-intensive procedures. Inkjet-based three-dimensional printing techniques have also been suggested due to their rapid and convenient operation in a scalable and programmable manner[23]. There have been several successful reports in patterning as-prepared synthetic melanin-like NPs[24] and catechol-conjugated biopolymers[25], but bottom-up fabrication (i.e., in situ growth from the surface in the desired shape) through inkjet printing is still a challenge, mainly due to the aforementioned adhesive property causing undesirable coating on all surfaces of apparatus (tubes, tips) contacting the ink and the technical difficulties in adjusting the coating speed while printing. More recently, approaches for the local initiation of the generation of melanin-like materials in aqueous solutions containing precursors have attracted great attention. Levkin's group developed a photo-patterning method by UV-induced local generation of reactive oxygen species (ROS) that triggers polydopamine synthesis in the local area[26]. Payne's group reported a redox-based direct writing method via an electrode pen, which can be applied to soft hydrogels[27]. d'Ischia's and Ball's groups reported localized coating on tyrosinase-containing hydrogel bead without spoilage of any material via leakage to solution[28]. Weil's group demonstrated that nanofabrication of poly-dopamine occurs site specifically on a catalytic DNAzyme motif[29,30]. In these approaches, only the locally oxidized precursors participate in further progressive assembly, forming melanin-like pigments.

Here, we report a novel method named PAINT, which stands for *Progressive Assembly on an Initiator-loaded Template*, for the site-specific fabrication/patterning of NIR-responsive melanin-like pigments. In conventional coating methods, the initiation of progressive assembly (i.e., the oxidation of phenolic and/or catecholic precursors) occurs in the coating solution such that all solution-exposed surfaces are covered by grown pigments (Fig. 1c, left). This is why conventional lithography techniques have been necessary to control the solution-exposed area for patterning. Different from previous approaches, we found that progressive assembly can be achieved in a spatially confined manner if it is simply initiated locally because the intrinsic underwater adhesion of in situ-generated oligomeric intermediates drives their localization to the nearby surface without diffusion to the solution. Therefore, in this PAINT approach, even if the entire surface is immersed in the coating solution, pigmentation occurs only in the specific area where the initiator is immobilized (Fig. 1c, right). As a proof-of-concept demonstration, we prepared a template by immobilizing a well-known enzyme, horseradish peroxidase (HRP), that mediates the oxidation of phenolic and/or catecholic precursors on a cellulose membrane. Next, we monitored the behavior of surface-

initiated pigment generation by simply immersing the templated membrane in aqueous media containing phenolic and catecholic precursors. Finally, one pigment most similar to natural melanin was selected, and its NIR-to-heat conversion efficiency was evaluated to validate its potential in biomedical applications.

## Results

### Solution-phase screening of precursor candidates for melanin-like pigment generation mediated by HRP

Prior to patterning by the PAINT approach, we first screened the precursor candidates for melanin-like pigment generation. Due to the uncontrollability of oligomeric intermediates, the chemical structure of the precursor is one of the only options available to alter the material properties of the formed pigment. Furthermore, this screening is necessary because enzymatic oxidation chosen for the initiation of PAINT is more precursor dependent than other chemical oxidations by sodium periodate, copper sulfate, and ammonium peroxodisulfate[31]. Figure 2a and Supplementary Figs. 1–5 represent a panel of color change and corresponding UV–vis-NIR absorbance of precursors in the absence and presence of HRP and hydrogen peroxide ($H_2O_2$). We tested 26 compounds, which were classified into 5 classes depending on their response to enzymatic conversion by HRP. The precursors belonging to classes 1–3 show distinct UV–vis-NIR spectra after enzymatic conversion, while others were rejected in further studies due to the intrinsic background signal of the precursor itself, autooxidation without enzymatic conversion, or a minimum response to HRP/$H_2O_2$ (Fig. 2a and Supplementary Figs. 4 and 5). The four precursors belonging to class 1 showed defined peaks corresponding to water-soluble single-molecule chromophores (Fig. 2b and Supplementary Fig. 1). The other five precursors belonging to class 2 resulted in nano-to-microscale particulates, detected by increased tunability at all tested wavelengths (Fig. 2c and Supplementary Fig. 2). However, the absorption characteristics of compounds in both classes 1 and 2 were far different from those of natural melanin. In fact, natural melanin possesses a characteristic broadband absorption in a range from ultraviolet to NIR regions due to the geometric packing of in situ-formed heterogeneous building blocks, which results in electronic transitions with charge transfer[32–35] (Fig. 2e). These absorption characteristics contribute to the multifunctionality of melanin, particularly harmful UV protection, visible indication/coloration, and thermal regulation through photothermal conversion (which are specifically important to cold-blooded animals) in natural species[36]. By mimicking these properties, melanin-like materials have been utilized in biomedical applications, such as photoprotection[37], photothermal and photodynamic therapies[38,39], and colorimetric detection/marking of biomarkers for disease diagnostics[40]. Finally, we found three precursors (belonging to class 3) as suitable candidates for PAINT, which were converted into pigments with broadband absorption characteristics resembling those of natural melanin after enzymatic conversion by HRP (Fig. 2d and Supplementary Fig. 3).

### The precursor-dependent behavior of surface-initiated pigment generation through PAINT

Next, we monitored the precursor-dependent behavior of surface-initiated pigment generation through PAINT by immersing the HRP-spotted cellulose membrane into aqueous media containing each precursor (Fig. 3a). In all tested precursors belonging to classes 1 to 3, the pigment was generated and simultaneously attached and/or adsorbed on the membrane where HRP was spotted. The background color surrounding the spots also appeared in several precursors, which was not only due to the intrinsic color of the adsorbed precursor but also the result of diffusion of surface-generated pigment into solution (Supplementary Figs. 6 and 7). The diameter of the resulting spots can be an indication of site-specific generation and/or localization of the formed pigment on the surface, which can be mainly affected by

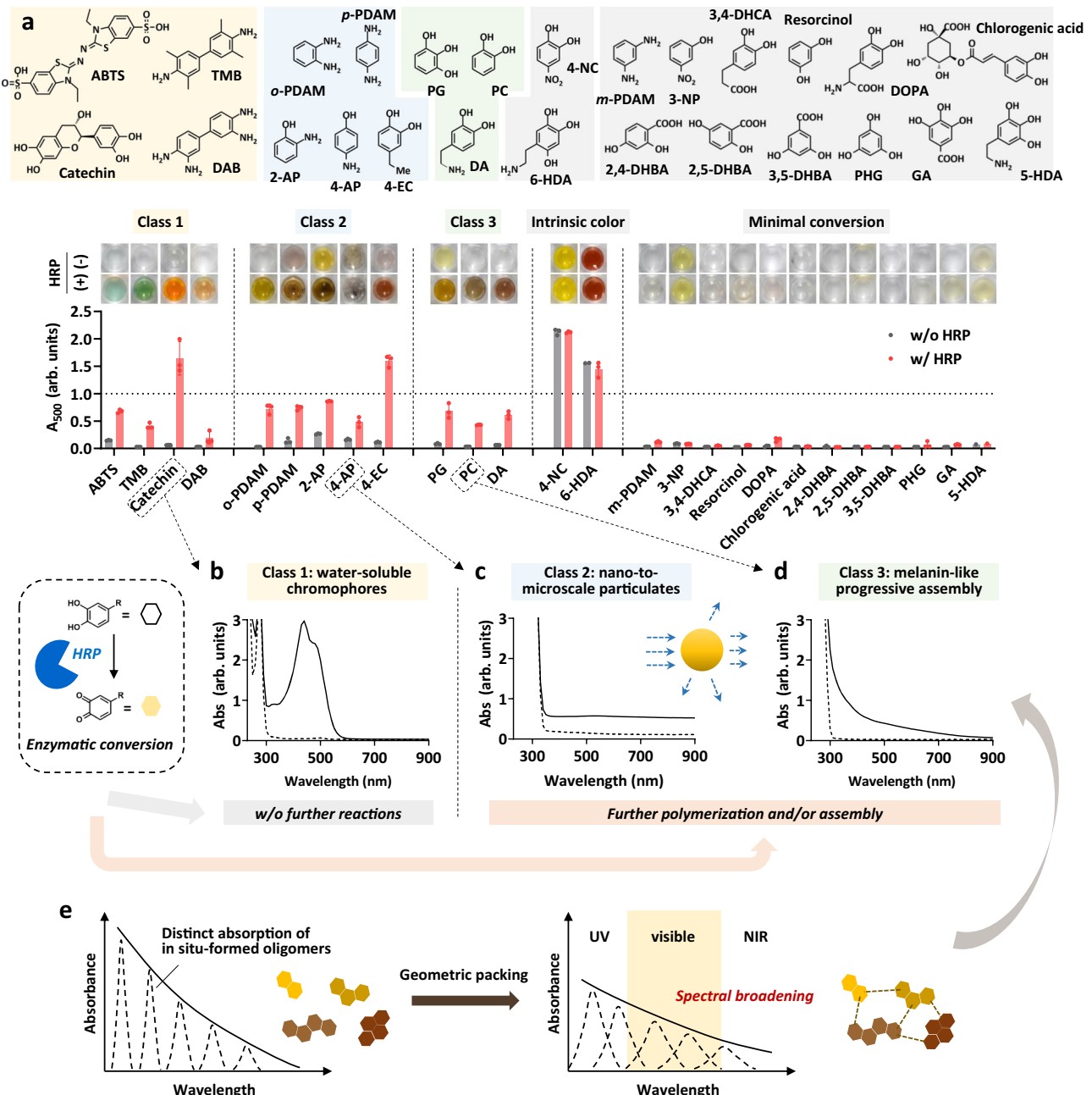

**Fig. 2 | Solution-phase screening of precursors for HRP-mediated pigmentation. a** A panel of color images and the absorbance at 500 nm of the tested precursors before and after treatment with HRP/$H_2O_2$ (HRP: 10 U/mL, $H_2O_2$: 20 mM) in solution for 60 min at 25 °C. Error bars represent the standard deviation (SD) of measurements on 3 independent samples. **b**–**d** Spectral change in representative precursor in three different classes upon oxidation by HRP/$H_2O_2$ (dotted line: intrinsic spectrum of precursor before oxidation, solid line: spectrum after HRP/$H_2O_2$ oxidation). **e** A schematic illustration indicating the characteristic spectral broadening of melanin-like pigment arising from the geometric packing of in situ-formed oligomers.

underwater adhesion. (Fig. 3b). The precursors belonging to class 3 (denoted as green color) resulting in melanin-like light broadband absorption, probably through progressive assembly, exhibited superior localization on the HRP-spotted region with the lowest spreading (100–127% diameter expansion). On the other hand, the precursors belonging to classes 1 and 2 (pink color denoting class 1 and blue color denoting class 2) showed a greater spreading (140–302% for class 1 and 150–321% for class 2 diameter expansion). Among classes 1 and 2, we further performed an additional classification and concluded that the chemical structure of precursors can be used to predict the adhesive properties of pigments instead of their type of spectra because polyphenolic compounds showed lower diameter expansion (140–162% diameter expansion) than non-phenolic compounds

(216–321% diameter expansion). This classification was also in line with previously reported results on the adhesive nature of phenolic materials[41].

The adhesive properties of pigments prepared from seven phenolic precursors were further comprehensively compared on a sacrificial gelatin hydrogel that readily melted at a high temperature above 60 °C (Fig. 3c). Similar to pigmentation on a two-dimensional cellulose membrane, the tested pigments were well generated locally on the HRP-printed area of the hydrogel (Fig. 3d, first row). Subsequently, it was observed that most of the pigments were dissolved as the supporting hydrogel melted, but the pigments prepared from 4-aminophenol (4-AP) and pyrocatechol (PC) as precursors retained the shape of the pigmented region (Fig. 3d, 2nd row). This was

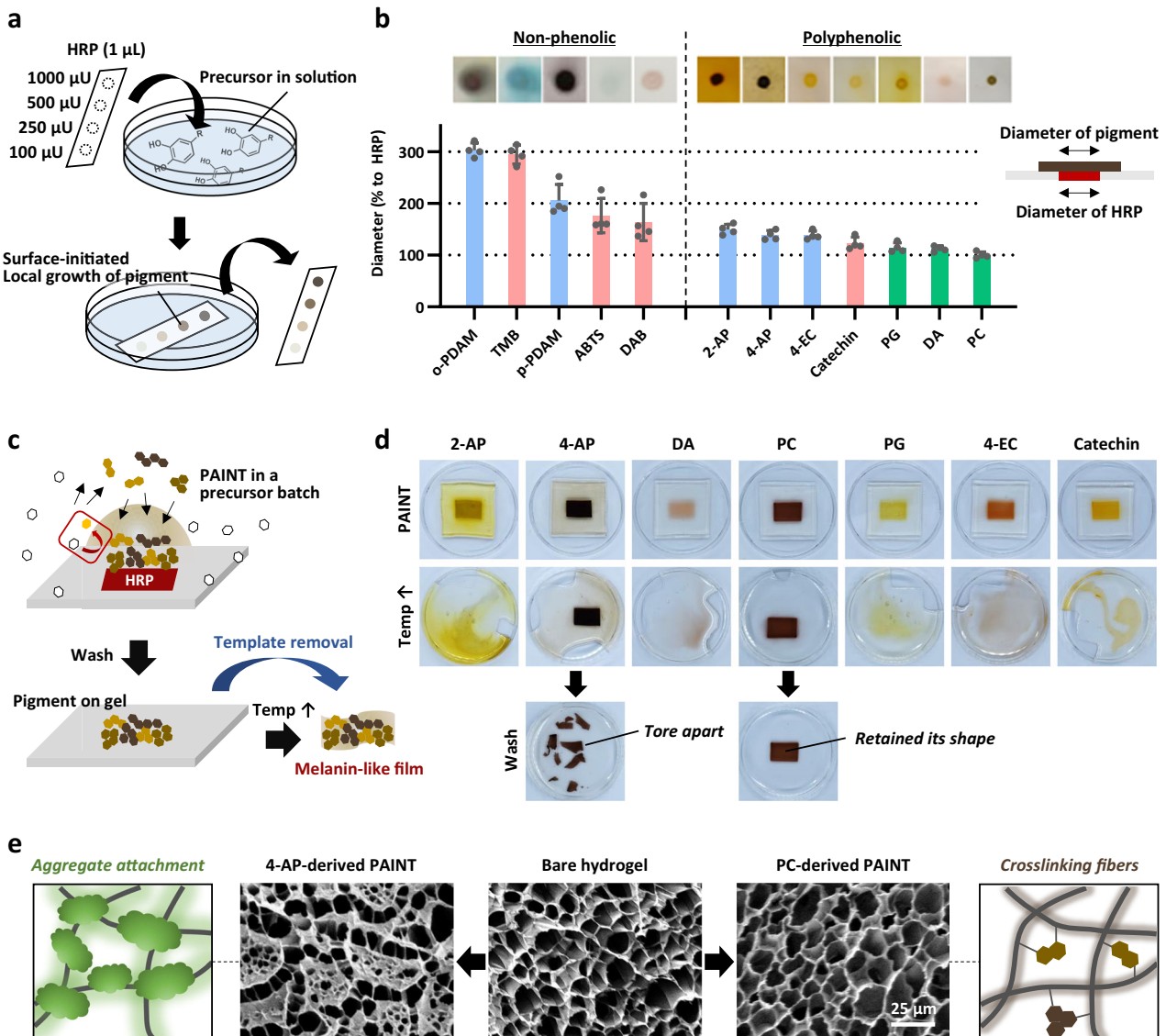

**Fig. 3 | Precursor-dependent behavior of surface-initiated pigment generation through the PAINT approach. a** Experimental scheme of pigmentation on the HRP-spotted cellulose membrane. Localized pigmentation occurs only on the HRP-spotted region on the membrane, and unreacted precursors remain in solution without color change. **b** Diameter expansion (%) of the pigment relative to the diameter of the HRP-spotted region for each precursor. This shows the localization of the surface-generated pigment. Each spot was generated from 1 μL of 1 U/mL HRP adsorbed on a cellulose membrane. Error bars represent the SD of measurements on 4 independent samples. **c** Experimental scheme of pigmentation on a sacrificial gelatin hydrogel followed by melting at 60 °C, resulting in a melanin-like free-standing film. **d** Images showing pigmented pattern disruption after dissolving the supporting hydrogel. **e** The surface morphology of the hydrogel before and after pigmentation indicating precursor dependency; aggregates were formed in the case of 4-AP-derived pigmentation, whereas molecular incorporation maintained the shape of the pigmented pattern without any microscopic changes in the fibrous structures of the hydrogel in the case of PC-derived pigmentation.

unexpected because neither the intensity nor the adhesive properties of pigments were correlated with this phenomenon: both the pigment prepared from 2-aminophenol (2-AP) (with higher intensity than the pigment prepared from PC in Fig. 3b) and the pigment prepared from pyrogallol (PG) (with higher localization than the pigment prepared from 4-AP in Fig. 3b) were all dissolved. Finally, vigorous washing distinguished the 4-AP-derived pigment from the PC-derived one; the pigment prepared from 4-AP tore apart, whereas that prepared from PC maintained its shape (Fig. 3d, third row). To investigate the differences between pigments prepared from 4-AP and PC, we further imaged the pigmented gel surface by scanning electron microscopy (SEM) after lyophilization (Fig. 3e). Interestingly, the hydrogel treated with 4-AP-derived pigment showed newly formed aggregates that almost blocked the pores (Fig. 3d, left). On the other hand, the surface of the hydrogel with PC-derived pigment had no microscopic

morphological changes from the untreated gel fibers; thus, it could be interpreted that the pigments were molecularly incorporated and/or smoothly coated as nano-thin layers on the polymeric network (Fig. 3d, right). In fact, catechol monomers are well-known crosslinkers of polymers, as found in the insect cuticle and squid beak sclerotization process[42,43], and their covalent reactions with surrounding nucleophiles, such as amines and thiols, have been utilized in synthesizing polymeric adhesives, hydrogels, and films[1,44–46]. In the same line with these reports, it is plausible that the PC-derived pigments were incorporated into the gelatin fibers, which strengthened the three-dimensional network of fibers even at the high temperature at which the bare gelatin gel melted. Finally, we selected a PC-based pigment for further PAINT demonstration because it showed the best localization on the HRP-immobilized surface without diffusion as well as high absorption in the visible region.

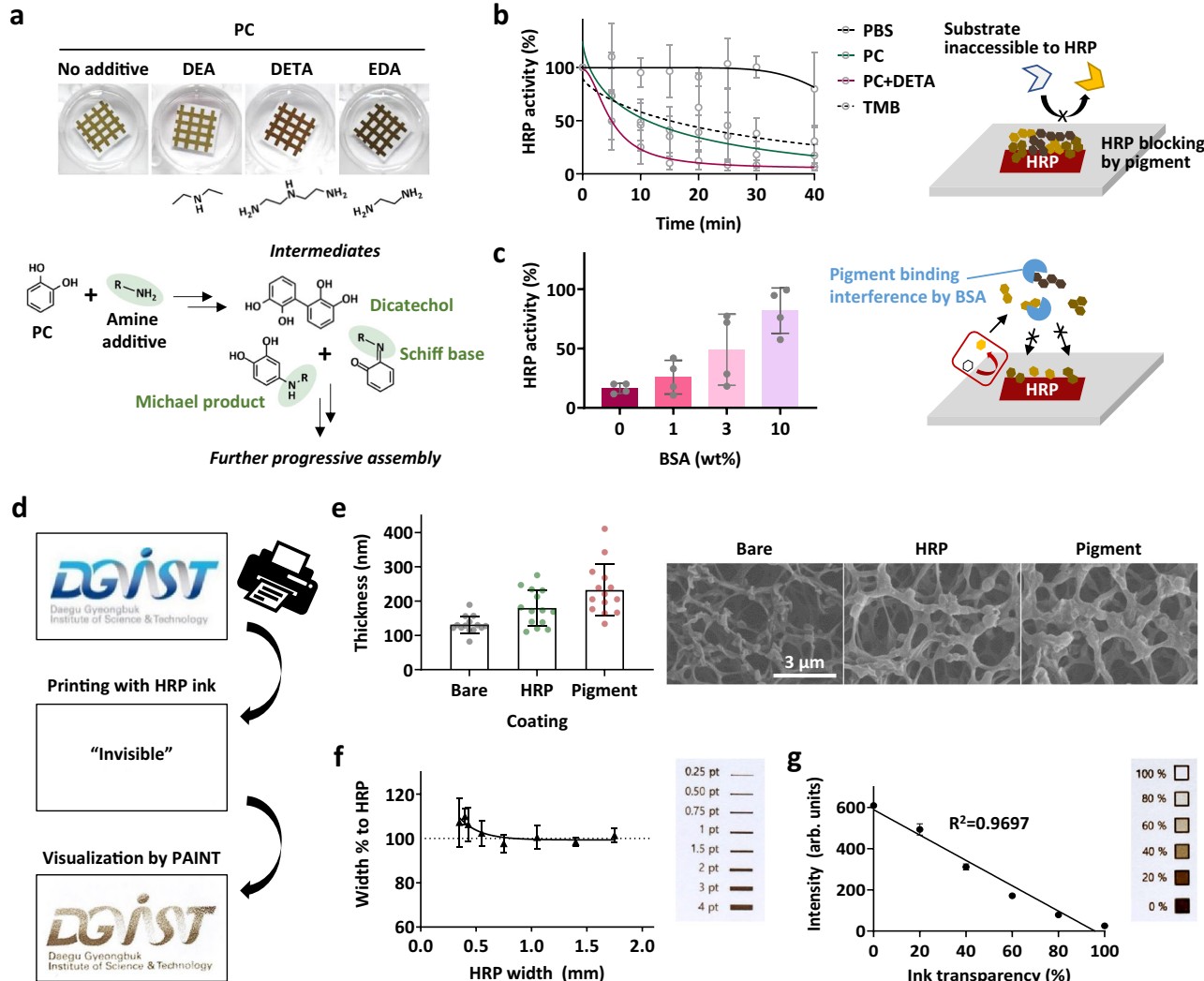

**Fig. 4 | Amine additive incorporation in PC-based PAINT. a** Pigment darkening effect of three different amine-containing additives and their plausible reactions with PC. **b** HRP activity monitoring as pigmentation occurring while immersing in aqueous solution of PC and DETA. Melanin-like attached pigment on HRP can act as a physical barrier preventing substrate access to HRP. TMB was used as a nonadhesive control for PC-based pigments. **c** Interference with pigment-derived HRP inactivation by BSA that can suppress the attachment of pigment to HRP. **d** A demonstration of pigmented pattern generation by inkjet-printed HRP. **e–g** In-depth characterization of the PC/DETA-based pigment fabricated on A4 paper with inkjet-printed HRP. **e** Thickness changes of fibrous structures of nitrocellulose membrane after HRP printing followed by pigmentation, observed by SEM imaging. **f** Pigment width expansion% from each inkjet-printed HRP with different widths ranging from 0.2 to 1.75 mm. **g** Quantitative analysis of the intensity of the fabricated pigment depending on the transparency of the HRP ink controlled by the software of the inkjet printer. Error bars in all graphs represent the SD of measurements on 3–5 independent samples unless sample size is not indicated.

## Incorporation of amine-containing additives during PC-based PAINT

Amine-containing molecules could participate in the PC-based PAINT. As shown in Fig. 4a, the two additives, diethylenetriamine (DETA) and ethylenediamine (EDA), containing primary amine groups, both resulted in pigment darkening, but diethylamine (DEA) did not. Several oligomeric intermediates were detected by HPLC-MS analysis of the solution-phase reaction of PC and amine additives with HRP (Supplementary Fig. 8 and Supplementary Table 1). In the case of the pigmentation of PC with DETA or EDA, the mass-to-charge ratio of intermediates detected after acidification could be calculated by the formation of multiple Schiff bases ejecting several water molecules. On the other hand, only single bond formations (possibly including Michael addition) were predicted in the case of the reaction between PC and DEA. In fact, it has been reported in many studies that both Michael addition and Schiff-base formation occur together upon reaction between catechol and amine groups[47–49]. Under our experimental conditions, only Schiff-base products were detected; however,

Michael addition cannot be completely rejected during the reactions of PC with DETA or EDA because aggregates were found to form in the solution (Supplementary Fig. 9). These aggregates were difficult to break down into individual intermediates and were thoroughly filtered out before HPLC–MS analysis. Therefore, it is still possible that the Michael-addition products could be formed but not detected because they were contained in the aggregates. In other words, the oligomeric intermediates detected by HPLC-MS only provide experimental support for progressive assembly, and do not serve as evidence for the dominance of Schiff-base formation in the reaction between PC and primary amines. The comparison of solution-synthesized pigment and the pigment fabricated on a gel by PAINT approach indicates that the detected intermediates were successfully localized on the surface (Supplementary Fig. 10). The incorporation of amine groups in PC-based pigmentation also affected the kinetics of the HRP inactivation occurring as in situ-generated pigment attached and physically prevented access of the substrate to HRP. As shown in Fig. 4b, the incorporation of DETA drastically enhanced the inactivation of HRP on the

surface during PC-based pigmentation; only 25.2% of the initial activity of HRP remained after 10 min of PC/DETA-based pigment growth, while 45.8% remained after PC-based pigment growth without DETA. The HRP inactivation kinetics of PC-based pigmentation without DETA were similar to those of the negative control, where the surface-bound HRP was treated with its well-known non-adhesive substrate, 3,3′,5,5′-Tetramethylbenzidine (TMB), indicating its minimum effect on the physical blocking of substrate access to HRP. Physical blocking after the attachment of PC/DETA-based pigment to HRP was further confirmed by adding BSA in the precursor solution, which can suppress the surface growth of pigments[50]. As shown in Fig. 4c, the activity of surface-bound HRP was less inactivated when a higher concentration of BSA was added to the precursor solution, as expected. In this case, the pigment that diffused into solution as a result of the interaction with BSA was detected (Supplementary Fig. 11).

Conventional inkjet printers can be used to prepare complex patterns by using HRP as an invisible ink on two-dimensional membranes, which can be readily visualized through PAINT method. In a demonstration, the logo of our institute was successfully reproduced with the melanin-like pigment, on which we could see the well-generated gradient colors and the high resolution of word letters on a 250 μm scale (Fig. 4d). The PC/DETA pigment fabrication on inkjet-printed HRP was further characterized in depth. HRP printed on conventional A4 paper increased the thickness of the fibrous structures by approximately 50 nm on average, and another increase of approximately 50 nm was observed after pigmentation (Fig. 4e). Consistent with the pigment generated on a hydrogel shown in Fig. 3e, the pigment was coated on the fibrous structures of the nitrocellulose membrane without any aggregates. The localization of pigment was again quantified by comparing the width of the fabricated pigment to that of the initially printed HRP; the spreading of pigment from the HRP pattern was less than 2% until as thin as 0.5 mm and increased to approximately 7–9% on narrower HRP patterns between 0.2 and 0.5 mm (Fig. 4f). PC-based pigment without DETA also showed the same trend for spatial expansion with respect to the HRP-printed region, although the color was paler than that with DETA (Supplementary Fig. 12). Finally, the linear relationship between the ink transparency adjusted by the software of the inkjet printer and the intensity of the resulting pigment was confirmed as expected (Fig. 4g). The printed PC/DETA pigment was stably maintained in 1× PBS (pH 7.4) at 37 °C for 7 days without any intensity change or remarkable leaching (Supplementary Fig. 13).

### Photothermal conversion activity of melanin-like pigment fabricated by PAINT for biomedical applications

As shown in Fig. 2d, melanin-like pigments can absorb NIR light. Therefore, we characterized the NIR-responsive properties of the PC/DETA-based pigment fabricated by PAINT to validate its feasibility in biomedical applications. Figure 5a shows the heat generation upon light irradiation (808 nm, 1 W/cm$^2$) on bare and pigmented A4 paper under wet and dry conditions. There was no temperature increase on either hydrated or dried bare A4 paper upon light irradiation, but the temperature of the pigmented region increased up to 136 °C in the dry state and 62 °C in the wet state after 1 min of light irradiation. The temperature change from the pigment-mediated photothermal conversion was reversible; the temperature increase by light irradiation (808 nm, 1 W/cm$^2$) and the decrease by rewetting was resumed up to five cycles without any damage to the pigmented patterns under wet conditions (Fig. 5b).

NIR-derived photothermal manipulation of cells has been used for the treatment of various diseases, mainly because of its minimal invasiveness[51]. Examples include cell ablation for the disinfection of medical devices and anti-cancer therapies, as well as tissue regeneration (e.g., nerve stimulation)[52–54]. In these applications, spatial control is crucial for minimizing side effects on the surrounding cells and tissues. Therefore, we envision that our melanin-like pigment fabrication by PAINT will be applied to produce implantable and/or wearable devices for localized cell and tissue stimulation in a spatially controlled manner. As a demonstration, photothermal ablation of mammalian cells seeded on the pigmented surface was performed (Fig. 5c). Both the bare and pigmented A4 paper showed good biocompatibility and cell adhesion comparable with conventional tissue culture plates (TCPs) (tested with NIH3T3) (Supplementary Fig. 14). Unexpectedly, the pigmented region showed significantly weaker background fluorescence. This may have been due to the prevention of dye adsorption and/or fluorescence quenching by the pigment on the A4 paper (red: ethidium homodimer-1 for dead cells; green: calcein-AM for live cells) (Supplementary Fig. 15). As the temperature increased by 1 W/cm$^2$ NIR irradiation at 808 nm for 10 min, most of the HeLa cells on the pigmented area became dead (viable cells per total cells on the surface: 98.4% before and 5.1% after NIR irradiation), as expected (Fig. 5d, e). Meanwhile, 86.3% of the cells on bare A4 paper still survived, although some cells died, probably due to heat transfer from the neighboring pigmented region in the same media during NIR irradiation.

In a second application, NIR-responsive pigmented hydrogels were employed as humidity-controlled soft actuators. Soft actuators can be used in various biomedical applications, such as soft tools for surgery, diagnosis and drug delivery, wearable assistive devices, prostheses, artificial organs and tissue-mimicking active simulators[55–57]. The heat generation from photothermal conversion on the pigmented region of hydrogels can induce local dehydration that alters the mechanical properties (Fig. 5f). As a demonstration, we fabricated a Janus-faced hydrogel actuator by coating one side of a polyacrylamide hydrogel with melanin-like pigment through PAINT (thickness of gel: 1 mm, thickness of pigment: 200 μm). As shown in Fig. 5g, h, the pigmented gel showed bending actuation as NIR irradiation time and power increased; the bending angle increased up to 47 ° within 5 min on 2 W/cm$^2$. The bent gel was reversibly stretched back by rehydrating the NIR-irradiated region, which was reversibly performed for up to 20 cycles without any damage to the gel (808 nm, 2 W/cm$^2$) (Fig. 5i).

## Discussion

In summary, we developed a new method named PAINT for site-specific fabrication of melanin-like pigments by revisiting the previously termed progressive assembly where heterogeneous intermediates are in situ formed and attached together to the surface. Due to the intrinsic underwater adhesive properties of the intermediates participating in the progressive assembly, we simply induced local pigment generation on an initiator-loaded region of the surface without any labor-intensive lithographic techniques. The spreading of pigment on the templating model initiator HRP was less than 2% until as thin as 0.5 mm, and interestingly, it was observed that the enzymatic activity of HRP diminished as the pigment was generated and covered it. The fabricated pigment through PAINT showed good biocompatibility and NIR-responsive properties, which can be useful in various biomedical applications. We selected HRP as our model initiator to demonstrate the PAINT approach. However, catalytic nanomaterials and enzyme-mimicking macromolecules like DNAzymes can also be adapted into PAINT as surface-loaded initiators. These variations in initiators further extend the range of applications of our PAINT method by possessing better stability than HRP for long-term storage and/or increasing the pigmentation speed. We expect that the PAINT method can be a useful toolkit that broadens the applications of melanin-inspired multifunctional materials on both two- and three-dimensional surfaces based on its convenience and versatility.

Most studies mimicking melanin have focused on achieving material properties, such as ROS scavenging and UV−visible−NIR adsorption. However, many of the underwater adhesive properties of melanin remain unknown. In melanin, particularly eumelanin, 5,6-dihydroxyindole (DHI) has been reported to play a crucial role in

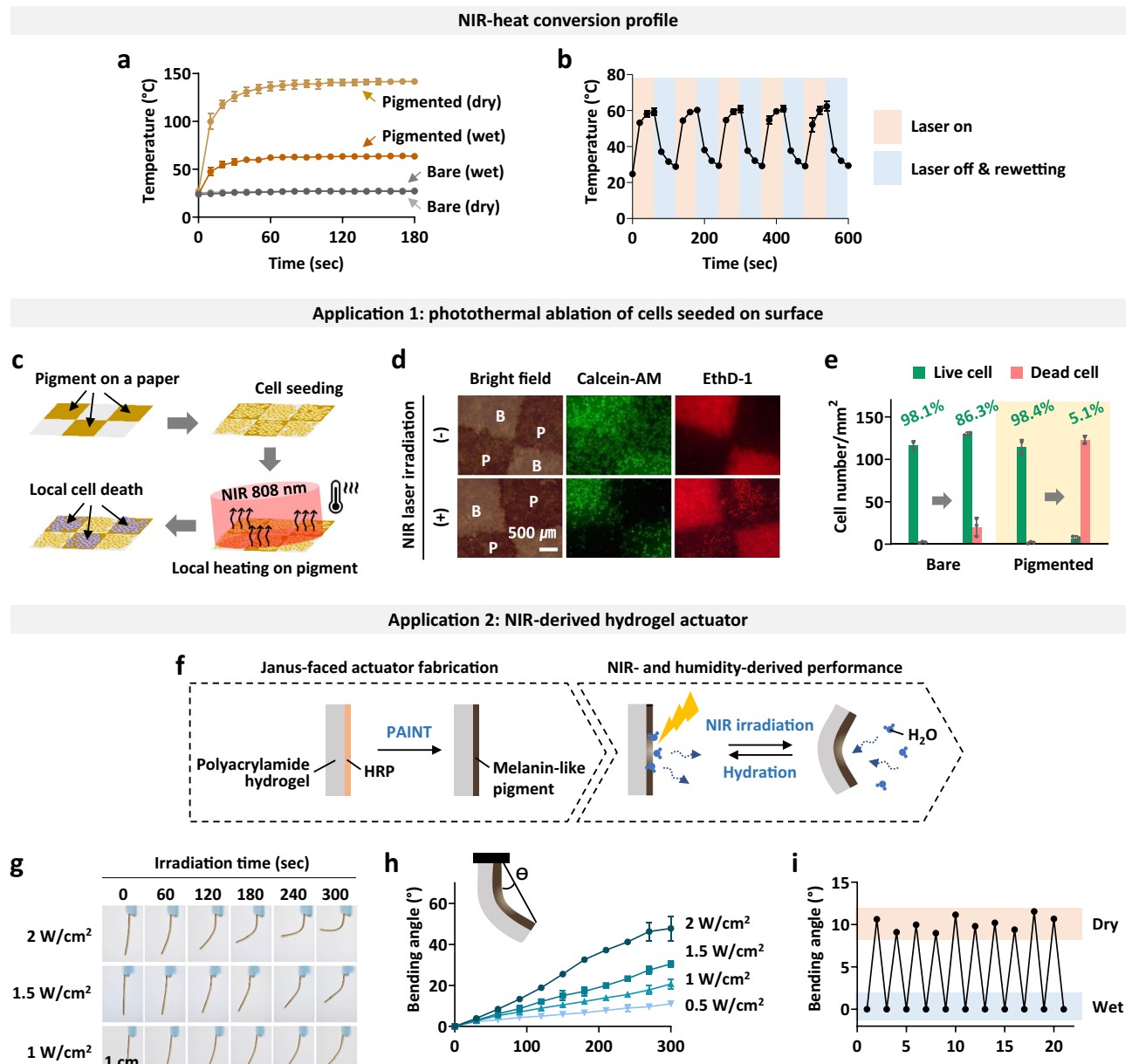

**Fig. 5 | Photothermal conversion properties of PC/DETA-based pigments for biomedical applications. a** Characterization of the heat generation efficiency of pigmented A4 paper upon NIR irradiation (808 nm, 1 W/cm²) in wet and dry conditions. **b** Reversible temperature changes by NIR irradiation and rehydration of the pigmented paper. **c–e** A demonstration of photothermal ablation of mammalian cells (NIH 3T3) seeded on pigmented paper. **c** A schematic illustration of the experimental procedure. **d** Cell viability on the bare (denoted as B in bright field image) and pigmented (denoted as P in bright field image) regions on paper upon NIR irradiation (808 nm, 1 W/cm²) (red: ethidium homodimer-1 staining for dead cells, green: calcein-AM staining for live cells). **e** Quantitative analysis of cell viability on bare and pigmented regions on paper based on the fluorescent images shown in **d**. **f–i** Demonstration of an NIR-responsive humidity-controlled soft actuator. **f** Schematic illustration of the fabrication procedure and actuation performance of the one-side pigmented hydrogel. **g** Bending actuation of the Janus-faced actuator as the NIR irradiation time and power change. **h** Quantitative analysis of the bending actuation based on the images shown in **g**. **i** Reversible bending of the fabricated actuator upon serial NIR irradiation and rehydration for up to 20 cycles (60 sec, 808 nm, 2 W/cm²). Error bars in all graphs represent the SD of measurements on 3–5 independent samples unless sample size is not indicated.

determining its properties. However, our study raises the possibility that DHI may not be important because PC-based precursors yield pigments with both UV–visible–NIR broadband adsorption and underwater adhesive properties. This requires further investigation.

## Methods
### Materials
PC, DETA, HRP, $H_2O_2$, and all precursors, *including dopamine hydrochloride*, were purchased from Sigma–Aldrich (USA). Purities of all purchased materials are shown in Supplementary Table 2. PBS buffer (1x) was purchased from Corning. The inkjet printer was purchased from Samsung (SL-J1560, KOREA). A4 paper was purchased from Quality (Thailand).

### HRP-mediated pigmentation in solution
The catechol precursor was first dissolved in 1× PBS, pH 7.4 (90 mM) containing 40 mM hydrogen peroxide. Subsequently, the precursor solution was mixed 1:1 with 1× PBS containing HRP (0.02 U/μL). The mixture was then incubated for 60 min at 25 °C and monitored by a

UV–vis spectrophotometer (Synergy H1 multimode microplate reader, Biotek).

## Pigmentation on the surface through the PAINT method

For inkjet printing of HRP on A4 paper, the conventional ink in the cartridge was first completely washed, and then the cartridge was filled with 10 U/mL of HRP in 1× PBS. Finally, the HRP pattern was printed on A4 paper in the same way as conventional inkjet printing. Inkjet-printed HRP on the transparent PET film was used as a stamp to achieve the transferred HRP pattern on the as-prepared gelatin hydrogel (10 wt % in 1× PBS) or polyacrylamide hydrogel. The HRP ink (100 to 1000 μU/μL in 1× PBS, 1 μL loaded) could also be directly spotted on a cellulose membrane. All types of HRP-loaded surfaces were immersed in 1× PBS containing 45 mM precursor and 8.16 mM hydrogen peroxide in the presence and absence of 22.5 mM DETA for 20 min at 25 °C and then washed with 1× PBS. Because hydrogen peroxide oxidizes the precursor (although the reaction rate is much slower than in the HRP-catalyzed system), hydrogen peroxide was added to the precursor solution immediately before dipping the HRP-loaded surfaces.

## HRP activity monitoring

HRP was first immobilized on a 96-well ELISA plate by incubation of 1 U/mL HRP in 1× PBS for overnight at 4 °C followed by washing with 1x PBS. Subsequently, 100 μL of precursor solution (mixture of 5 mM PC and 10 mM $H_2O_2$, in the presence or absence of 5 mM DETA) was added to each well and then incubated at 37 °C. At each time point, the precursor solution was removed, and each well was washed with 1× PBS. Finally, 100 μL of a 9:1 mixture of 1× PBS containing 20 mM $H_2O_2$ and 1 mg/mL TMB in DMSO was added to each well and incubated for 60 min, and finally, the absorbance of the reacted solution at 650 nm was measured. As a comparison, a 9:1 mixture of $H_2O_2$ and TMB solutions was used instead of the precursor solution in pigment generation.

## Characterization

Morphological changes in the fibrous substrates were observed by field emission scanning electron microscopy (FE-SEM, Hitachi, S-4800). Samples were precoated with platinum (3 nm) to prevent surface charging during imaging. UV–Vis-NIR spectra were obtained using a spectrophotometer (Synergy H1 plate reader, BioTek). HPLC–MS analysis was carried out using a liquid chromatograph (6420, Agilent Technologies) equipped with a tender mass spectrometer. A gradient mixture of water and acetonitrile containing 0.1% TFA (the acetonitrile percentage was gradually increased from 10 to 90% in 10 min) was used as the mobile phase. A C18 column was used, the flow rate was 0.4 mL/min, and the sample injection volume was 5 μL.

## Biocompatibility test and photothermal ablation of cells seeded on pigment

NIH3T3 cells were cultured in DMEM with 10% FBS and 1% antibiotics at 5% $CO_2$ with humidity at 37 °C. For the culture of HeLa cells, MEM was used instead of DMEM. Prior to cell seeding, the pigmented A4 paper was washed with 70% ethanol in water and air-dried for sterilization. For biocompatibility test, NIH3T3 cells ($6 \times 10^4$ cells) were seeded on each well of an 8-well plate containing 0.7 $cm^2$ paper and cultured for 24 h in a cell culture incubator under 5% $CO_2$ with humidity at 37 °C. Subsequently, cells on the paper were stained with ethidium homodimer-1 and calcein-AM following the manufacturer's guidelines for the commercial LIVE/DEAD™ viability/cytotoxicity kit from Invitrogen™. For the photothermal ablation test, HeLa cells ($5 \times 10^4$ cells) were seeded on each well of an 8-well plate containing a 0.7 $cm^2$ sized paper having both bare and pigmented regions together. After 24 h of culture, cells on the paper were washed with 1× PBS and irradiated with NIR light (808 nm, 1 W/$cm^2$) in 1× PBS for 10 min and

incubated for 5 h in cell culture media. Finally, the viability of cells remaining on the paper was evaluated by the aforementioned LIVE/DEAD™ viability/cytotoxicity kit from Invitrogen™. Images were analyzed using ImageJ software to quantify both the live and dead cells attached to the surface.

## NIR-derived actuation of Janus-faced hydrogel

An polyacrylamide hydrogel was prepared by polymerization of acrylamide in the presence of APS and TEMED. One side of the prepared gel was pigmented by HRP stamping followed by immersion in precursor solution containing PC, DETA, and $H_2O_2$ as mentioned above. The pigmented side of the sample was irradiated with NIR light (808 nm, 0.5–2 W/$cm^2$) at a 5 cm distance for 5 min. The temperature change during light irradiation was monitored by using a thermal imaging camera (Flirone pro, USA), and the bending of the gel was imaged by digital camera (Nikon). Reversibility in bending actuation was investigated by cycling the laser on (808 nm, 2 W/$cm^2$) and off every 60 s. During off the light, the gel was briefly immersed in water at 40 °C for rewetting.

## Reporting summary

Further information on research design is available in the Nature Portfolio Reporting Summary linked to this article.

## Data availability

All data needed to support the conclusions in the paper are present in the paper and/or the Supplementary Information. The data that support the findings of this study are openly available in figshare at https://doi.org/10.6084/m9.figshare.22573771. Source data are provided with this paper.

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

## Acknowledgements

This work was supported by the National Research Foundation of Korea (NRF) grant funded by the Government of Korea (MSIT), the Basic Research Program in Science and Engineering (NRF-2020R1C1C1010700; to S.H.), and the Engineering Research Center (ERC) program (NRF-2018R1A5A1025511; to S.H.). S.H. also gratefully acknowledges support by the DGIST R&D programs of the Ministry of Science and ICT (21-HRHR-02 and 22-SENS2-01).

## Author contributions

H.J. and S.H. conceived the project and designed the experiments. H.J., J.L., S.K., and H.M. conducted experiments and data analysis. H.J. and S.H. discussed the results and wrote the manuscript with help from all authors.

## Competing interests

The authors declare no competing interests.
