## [Peer Review File · Nature Communications]

Site-specific fabrication of a melanin-like pigment through spatially confined progressive assembly on an initiator-loaded templateREVIEWER COMMENTS

Reviewer #1 (Remarks to the Author):

The manuscript by Jeong et al. studied site-specific adhesion of melanin-like pigment by HRP loaded template. The progressive assembly of melanin-like pigment is an interesting topic to study since it closely related to human physiological activities. The results show that the local progressive assembly could be induced on the given surface pretreated with initiators and could undergo NIR-to-heat conversion. However, many important issues are not well addressed in the present manuscript. The detailed comments are listed below.

1. The significance of site-specific patterning of melanin should be clearly clarified. What's the potential application in biomedicine? If you want to demonstrate cell ablation, what's the practical scenario it might relate to ?
2. Fig.3B: The underwater adhesion was quantified by diameter expansion. I don't think it is appropriate. The adhesion also relates to pigment intensity. In addition, is there any oxidized polyphenol in solution?
3. Fig.4A: what's the purpose for pigment darkening? You should explain it in the manuscript.
4. Fig.4D: there is a brown background for the decrypted pattern. Does it suggest diffused adhesion?
5. Fig.4F: you should study the resolution of PC-based PAINT.
6. The stability of assembled pigment on substrates should be studied. It will be of important if you want to quantify assembled melanin. Will the intensity change with time?
7. Fig.5D, I don't think it is appropriate to call the HRP coated hydrogel as Janus-faced pigmented hydrogel.
8. The soft actuator demonstration is not a sophisticated biomedical application.
9. There are no definitions for many abbreviates, such as OHP, TMB, DEA.

Reviewer #2 (Remarks to the Author):

In their manuscript titled "Site-specific fabrication of a melanin-like pigment through underwater adhesion-derived, spatially confined progressive assembly on an initiator-loaded template" Jeong et al. describe a method for the local formation of melanin-like pigments using immobilized enzymes, due to local adhesion of the species formed during monomer oxidation. The method is ingenious and can be easily applied to a number of different surfaces. This work is of great interest to the community and it was a true pleasure to read this excellent manuscript. I recommend publishing after some minor changes are made.

- 1) In the introduction, the authors describe methods for the local preparation of melanin-like materials but do not cite approaches using electropolymerization.
- 2) I am missing an explanation, for example in the experimental section how the values for the x- and y-axes in Figure 3A were obtained. How is the intensity of a pigment measured? What are the corresponding units?
- 3) I do not follow the nomenclature for the different species that were identified in Figure S7. Can the

authors reconsider their presentation and discussion of this data?

4) I could not find information in the experimental section on how the height of the samples after HRP deposition and pigment formation was determined.

5) Figure S9 and discussion in lines 234-235. The authors attribute the loss of fluorescent signal to a non-attachment of dye, but fluorescence quenching by melanin-like materials may also play a role here. The authors should discuss this possibility.

6) I disagree with the argument (lines 266-268) that melanin has rarely been mimicked. Plenty of literature on melanin-like materials exist. Instead, the authors should highlight their own achievement of highly localized synthesis of melanin-like materials.

7) References 27 and 28 (line 82): please check the name of the group.

Reviewer #3 (Remarks to the Author):

This paper by Jeong et al. is a further contribution by this research group to the exploitation of the adhesive properties of pigments obtained from dopamine or related compounds. In this study in particular a drawback of polydopamine related coatings that is the low site specific adhesion is addressed through a proof of concept approach in which an initiator, actually an oxidizing enzyme, is adsorbed on papers or hydrogels in order to promote pigment formation only on the areas where the initiator/oxidant has been deposited. After a selection of the best performing precursors the method is applied for writing on paper or hydrogels by ink jet distribution of the initiator followed by exposure to the solution of the precursor and hydrogen peroxide. Finally, two applications are reported to exemplify the potential of the so-called PAINT method.

The idea underlying this work may be of interest. As the authors state in the introduction this is not completely new as other related work have appeared though the way the site specificity is achieved is somewhat different. Beside the papers cited please consider also *Int. J. Mol. Sci.* 2020, 21, 4873 where tyrosinase-loaded alginate spheres, or films deposited on glass or polyethylene allowed PDA pigment formation from tyramine (converted to dopamine by tyrosinase) or also *Biofabrication* 2020, 12, 035009 for a study of the printability of metal ion crosslinked PEG-catechol based inks

There are several issues that should be considered by the authors

Introduction :

Line 54 and ff.: while the notion that melanin grows on a protein template during melanosome maturation from stage I to IV before their transfer to keratinocytes is well recognized, the idea that this is an example of underwater adhesion is not really supported by solid evidence. Tight adsorption on the protein matrix or even covalent linking of some pigment units to the protein nucleophilic residues are also viable options. Ref 17 and 18 quoted seem not appropriate to melanogenesis in nature as they refer to a model system of amyloid fibers and polydopamine, or cell free melanogenesis in fungi.

Line 92-93: Actually the oligomer intermediates arising from oxidation of the precursors are not identified nor characterized in this work so their intrinsic underwater adhesion is just a hypothesis. I

recommend to caution this sentence and just report what has been demonstrated.

Results:

It is not clear why the authors that intended to investigate melanin like pigments select in their preliminary screening a number of substrates that can not give rise to melanin type pigments like a series of anilines, resorcinols, hydroxybenzoic acids. The oxidation chemistry of these compounds is well known and most the observations reported could be predicted based on literature data or even chemical reasoning of the structure of these potential precursors. Also, there are some intriguing finding e.g. 6-hydroxydopamine appears red in color before oxidation whereas it should exhibit absorption only in the UV region (it seems it is already oxidized before exposure to the oxidant). It is very difficult to follow the text describing figure 2 as the abbreviations of the compounds tested are not reported under the structures (available only in the SI figures).

Line 182 and ff :The role of amine compounds in pigment darkening. The author state “we isolated a few reactions based on several identified water soluble intermediates (in Fig S7)” the sense of this sentence is quite obscure so as data reported in table of fig S7. Please show some representative structures in the main text. Also, a darkening that is a bathochromic shift is more likely the result of the addition of primary amino groups to catechols (as is the case of PC) rather than Schiff base formation. Possibly oxidation products of aminocatechols contribute to such darkening process as well. The meaning of sentences lines 188-193 is not clear: why the products resulting from Michael addition should not be detected by HPLC analysis? under the acid elutographic conditions the adducts should be protonated Line 203 why the author expected that BSA should be able to “abstract “in situ-grown oligomers before attachment to HRP”. Any evidence from previous studies?

Line 206 and ff figure 4: it is rather intriguing to see the arrow correlating the first panel in fig 4D to the second one with the label encrypted, while it appears that the experiment described is just decrypting the invisible logo after exposure to the catechol solution

A general consideration that the author should address and comment is the actual stability of the solutions of the catechols containing hydrogen peroxide which are used for dipping the HRP printed paper/hydrogel

Discussion: I would recommend to comment also on the difficulties intrinsic in the PAINT method, to caution the versatility of the proposed methodology. Also the view that melanin is just an example of a catechol material is rather simplistic as in the animal kingdom melanins are indeed more complex as the author state in the introduction and such complexity is the result of the chemistry of 5,6-dihydroxyindole biosynthetic precursors not any catechol system

REVIEWER COMMENTS

Reviewer #1 (Remarks to the Author):

The manuscript by Jeong et al. studied site-specific adhesion of melanin-like pigment by HRP loaded template. The progressive assembly of melanin-like pigment is an interesting topic to study since it closely related to human physiological activities. The results show that the local progressive assembly could be induced on the given surface pretreated with initiators and could undergo NIR-to-heat conversion. However, many important issues are not well addressed in the present manuscript. The detailed comments are listed below.

1. The significance of site-specific patterning of melanin should be clearly clarified. What's the potential application in biomedicine? If you want to demonstrate cell ablation, what's the practical scenario it might relate to?

As mentioned in the manuscript (page 4, line 124-132), melanin is photo-responsive with broadband absorption across the UV, visible and IR regions. This gives melanin multifunctionality in nature, ranging from UV protection, coloration, and thermal regulation through photothermal conversion (which is particularly important for cold-blooded animals). Therefore, melanin-mimicking materials could be used in biomedical applications such as photoprotection of skin, photothermal and photodynamic therapies, and colorimetric detection and marking of biomarkers for disease diagnosis.

Spatial control is important in disease treatment to minimize side effects to the surrounding cells and tissues. However, intrinsic universal adhesion is a drawback of on-demand nano-to-micro fabrication of current melanin-mimicking materials. Therefore, we envision that our new strategy for site-specific patterning will broaden the biomedical applications of multifunctional melanin-mimicking materials, in particular photo-responsive theranostics.

In this regard, we selected photothermal cell ablation to demonstrate the applicability of our site-specific patterning strategy in biomedical applications where spatial control is important. NIR-light-based photothermal manipulation of cells has the advantage of minimum invasiveness and has been used for cell ablation for the disinfection of medical devices and anti-cancer therapies, as well as tissue regeneration (e.g., nerve stimulation). As mentioned above, localized stimulation is crucial in these applications to minimize side effects on the surrounding cells and tissues. The results shown in **Figure 5C-5E** highlight the possibility of localized cell manipulation by site-specific patterning and fabrication of melanin-mimicking materials. This can be expanded to implantable and/or wearable soft devices for anti-cancer therapies, neural tissue regeneration, and other applications.

We have revised the manuscript to clarify the significance of site-specific patterning in biomedical applications as follows:

[Before, page 6, line 241-243] Cell ablation through NIR light can be useful in the disinfection of medical devices (such as implantable devices) without causing any damage and in cancer therapy due to the relatively high penetration depth in biological tissues and low energy of NIR light [45].

[After, page 6, line 243-249] NIR-derived photothermal manipulation of cells has been used for the treatment of various diseases, mainly because of its minimal invasiveness [51]. Examples include cell ablation for the disinfection of medical devices and anti-cancer therapies, as well as tissue regeneration (e.g., nerve stimulation) [52-54]. In these applications, spatial control is crucial for minimizing side effects on the surrounding cells and tissues. Therefore, we envision that our melanin-like pigment fabrication by PAINT will be applied to produce implantable and/or wearable devices for localized cell and tissue stimulation in a spatially controlled manner.

2. Fig.3B: The underwater adhesion was quantified by diameter expansion. I don't think it is appropriate. The adhesion also relates to pigment intensity. In addition, is there any oxidized polyphenol in solution?

We agree with the reviewer's comment that underwater adhesion does not linearly correlate with the expansion of the pigment diameter, and that it is also related to the pigment intensity.

We found in our additional experiment (**Figure S7**) that some of the pigment diffused into the surrounding solution during surface-initiated generation (for example, TMB or o-PDAM). We believe this observed diffusion into the solution reduced the pigment intensity on the surface. However, in case of the PC, there was no pigment diffused into solution, which supports our PAINT mechanism.

To avoid confusing readers, we now explain the diameter expansion in **Figure 3B** as site-specific generation and/or localization of the formed pigment on the surface, which can be mainly affected by underwater adhesion. In addition, we have removed the intensity term from **Figure 3B** because it is not orthogonal to the adhesion. The revised figure and corresponding text in the manuscript are shown below.

[**Figure 3B, before**]

(B) Comparison of underwater adhesion (quantified as diameter expansion% from HRP-immobilized spot) and turnover rate (corresponding to pigment intensity in grayscale) of on-surface generated pigment from each precursor.

[**Figure 3B, after**]

(B) Diameter expansion (%) of the pigment relative to the diameter of the HRP-spotted region for each precursor. This shows the localization of the surface-generated pigment.

[**Before, page 4, line 135-137**] The background color surrounding the spots also appeared in several precursors, which was due to the intrinsic color of the adsorbed precursor and not from the HRP-mediated reaction (**Figure S6**). However, the diameter of the resulting spots varied due to the differences in the underwater adhesive properties of the on-surface synthesized pigment (**Figure 3B**).

[**After, page 4, line 138-143**] The background color surrounding the spots also appeared in several precursors, which was not only due to the intrinsic color of the adsorbed precursor but also the result of diffusion of surface-generated pigment into solution (**Figure S6-S7**). The diameter of the resulting spots can be an indication of site-specific generation and/or localization of the formed pigment on the surface, which can be mainly affected by underwater adhesion. (**Figure 3B**).

[**Removed, page 4, line 149-154 in the original manuscript**] In addition, the enzymatic turnover rate, primarily determining the intensity of the pigment (**Figure 3B**, y-value), could also be an indirect factor affecting pigment adhesion/localization. For example, the well-known adhesive material, pigment prepared from DA (i.e., polydopamine), exhibited unexpected spreading, possibly due to the relatively low enzymatic turnover rate, which results in a relatively low concentration of oligomers, making them less likely to adhere to each other.

[Newly added data in Figure S7]

Figure S7. UV–visible spectra of diffused pigment in solution during surface-initiated pigmentation of various precursors (red: incubation w/o HRP-spotted membrane, green: incubation with HRP-spotted membrane).

3. Fig.4A: what's the purpose for pigment darkening? You should explain it in the manuscript.

Pigment darkening by amine additives was an incidental finding, not one of our goals. The reason why we investigated the effect of amine-containing molecules on pigmentation is that the covalent reactions between catechol and nucleophiles, such as amines and thiols, have been used in the synthesis of polymeric adhesives, hydrogels, and films, as mentioned in the manuscript (page 5, line 173-176). Unexpectedly, the incorporation of amine additives drastically enhanced the inactivation of surface-bound initiator (HRP) by physical blocking after pigmentation in PC-based system (**Figure 4B-4C**).

4. Fig.4D: there is a brown background for the decrypted pattern. Does it suggest diffused adhesion?

We don't believe that diffusion occurred in solution because we did not detect oxidized PC in the solution (shown in the newly added **Figure S7** following reviewer 1's comment 2).

We repeated the experiment and took a photograph with bare paper without any HRP pattern to confirm that there was no difference in the background, as shown in right figure. We think the slightly darker background of the original **Figure 4D** might be due to environmental factors, such as the brightness and/or camera settings when taking the photograph of the wet paper. We have replaced the original **Figure 4D** with the result of the repeated experiment.

5. Fig.4F: you should study the resolution of PC-based PAINT.

We performed an additional experiment to study the resolution of the PC-based pigment and added the results to the supporting information instead of the original **Figure 4F**, because the purpose of **Figure 4D-4G** was to demonstrate inkjet printing with PC/DETA-based pigment. As shown in the newly added **Figure S12**, the spatial expansion of PC- and PC/DETA-based pigments was similar, although the color was paler without DETA.

[Newly added Figure S12]

Figure S12. Width expansion (%) of PC-based pigment from each inkjet-printed HRP pattern with widths ranging from 0.2 to 1.75 mm.

[Added, page 6, line 227-229] PC-based pigment without DETA also showed the same trend for spatial expansion with respect to the HRP-printed region, although the color was paler than that with DETA (Figure S12).

6. The stability of assembled pigment on substrates should be studied. It will be of important if you want to quantify assembled melanin. Will the intensity change with time?

We performed an additional experiment following the reviewer’s suggestion. As shown in the newly added **Figure S13A**, the intensity of the pigment did not change until day 7 of incubation in 1×PBS (pH 7.4) at 37 °C. In addition, there was no remarkable leaching of compounds into the solution, as confirmed by UV–visible spectroscopy (Figure S13B).

[Newly added Figure S13]

Figure S13. Intensity monitoring of PC/DETA pigment (A) and the detection of leachable products formed (B) during the incubation of printed PC/DETA pigment maintained in 1×PBS (pH 7.4) at 37 °C for 7 days.

[Added, page 6, line 231-232] The printed PC/DETA pigment was stably maintained in 1×PBS (pH 7.4) at 37 °C for 7 days without any intensity change or remarkable leaching (Figure S13).

7. Fig.5D, I don’t think it is appropriate to call the HRP coated hydrogel as Janus-faced pigmented hydrogel.

We meant to use the term “Janus-faced” for the one-side pigmented hydrogel after PAINT application on the HRP-coated hydrogel. **Figure 5F** shows the fabrication procedure of this hydrogel followed by

its actuation performance. We agree that the term “Janus-faced pigmented hydrogel” is inappropriate. Therefore, we now only use the terms “Janus-faced hydrogel” and “Janus-faced actuator”. Accordingly, we revised the figure legend not to confuse readers as detailed below.

[Before, page 18, legend of Figure 5] (F-I) A demonstration of an NIR-responsive humidity-controlled soft actuator generated by a Janus-faced pigmented hydrogel. (F) A schematic illustration of the fabrication and actuation performance of the fabricated hydrogel.

[After, page 20, legend of Figure 5] (F-I) Demonstration of an NIR-responsive humidity-controlled soft actuator. (F) Schematic illustration of the fabrication procedure and actuation performance of the one-side pigmented hydrogel.

8. The soft actuator demonstration is not a sophisticated biomedical application.

The authors do not agree with the reviewer’s comment. Soft actuators are regarded as promising for soft robotics in biomedical applications, such as soft tools for surgery, diagnosis and drug delivery, wearable assistive devices, prostheses, artificial organs and tissue-mimicking active simulators for training and biomechanical studies (M. Cianchetti, et al., *Nat Rev Mater*, 3, 143-153, 2018). The biocompatibility and photothermal actuation performance that we showed in this study further demonstrate the feasibility of adapting such systems in the future for the aforementioned biomedical applications.

We added examples (with references) of possible biomedical applications of our actuating hydrogel system to the text, as described below.

[Added, page 7, line 262-264] Soft actuators can be used in various biomedical applications, such as soft tools for surgery, diagnosis and drug delivery, wearable assistive devices, prostheses, artificial organs and tissue-mimicking active simulators [55-57].

[Added reference] [57] Cianchetti, M., Laschi, C., Mencias, A., Dario, P. Biomedical applications of soft robotics. *Nat. Rev. Mater.* 3, 143-153 (2018).

9. There are no definitions for many abbreviates, such as OHP, TMB, DEA.

We apologize for this oversight. We have carefully checked all abbreviations and acronyms, and added all missing definitions.

Reviewer #2 (Remarks to the Author):

In their manuscript titled “Site-specific fabrication of a melanin-like pigment through underwater adhesion-derived, spatially confined progressive assembly on an initiator-loaded template” Jeong et al. describe a method for the local formation of melanin-like pigments using immobilized enzymes, due to local adhesion of the species formed during monomer oxidation. The method is ingenious and can be easily applied to a number of different surfaces. This work is of great interest to the community and it was a true pleasure to read this excellent manuscript. I recommend publishing after some minor changes are made.

1) In the introduction, the authors describe methods for the local preparation of melanin-like materials but do not cite approaches using electropolymerization.

The electropolymerization method was referred to on page 3, line 80-81 in the original manuscript:

[Now on page 3, line 81-82 in the revised manuscript] Payne’s group reported a redox-based direct writing method via an electrode pen, which can be applied to soft hydrogels [26].

2) I am missing an explanation, for example in the experimental section how the values for the x- and y-axes in Figure 3A were obtained. How is the intensity of a pigment measured? What are the corresponding units?

We apologize for this confusion. The intensity (y-axis) was quantified using ImageJ software after converting the image to grayscale. The underwater adhesion (x-axis) was calculated from the diameter expansion (%) relative to the diameter of the HRP-spotted region (i.e., the ratio between the diameter of the generated pigment to that of the pretreated HRP on the surface, as shown in the inset illustration in the revised **Figure 3B**).

We revised **Figure 3B** based on reviewer 1's comment 2, which raised the point that the diameter expansion (x-axis) and intensity (y-axis) are not orthogonal to each other. Accordingly, the intensity term has now been removed from **Figure 3B**, and the explanation of the data (including the calculation of the diameter expansion) has been clarified in the figure legend, as shown below.

[Figure 3B, before]

(B) Comparison of underwater adhesion (quantified as diameter expansion% from HRP-immobilized spot) and turnover rate (corresponding to pigment intensity in grayscale) of on-surface generated pigment from each precursor.

[Figure 3B, after]

(B) Diameter expansion (%) of the pigment relative to the diameter of the HRP-spotted region for each precursor. This shows the localization of the surface-generated pigment.

3) I do not follow the nomenclature for the different species that were identified in Figure S7. Can the authors reconsider their presentation and discussion of this data?

Because different structures may have the same mass, we were careful not to draw any conclusions about the existence of specific structures based only on the detected mass values. We revised **Figure S8** with plausible structures for better understanding to readers and edited text as detailed below.

[Before, supporting information, Figure S7]

R-NH ₂	Species	Elution time	m/z	PC	R-NH ₂	- [H ₂ O]	- [C-C]
DETA	1	1.5 min	366	2	2	3	3
	2	2.0 min	366	2	2	3	3
	3	2.7 min	566	4	2	4	4
EDA	4	3.6 min	480	4	2	4	4
DEA	5	1.7 min	358	2	2	-	3

[After, supporting information, Figure S8]

Figure S8. HPLC–MS analysis of water-soluble intermediates identified from the PC-based pigmentation with three different amine additives.

[Before, page 5, line 184-186] Further studies via HPLC–MS analysis were performed to identify the key reactions between PC and these amine additives, and we isolated a few reactions based on several identified water-soluble intermediates (Figure S7).

[After, page 5, line 184-186] Several oligomeric intermediates were detected by HPLC-MS analysis of the solution-phase reaction of PC and amine additives with HRP (Figure S8).

4) I could not find information in the experimental section on how the height of the samples after HRP deposition and pigment formation was determined.

Instead of the height of the pigment, we measured the thickness of the fibers after pigment deposition observed by scanning electron microscopy, as shown in **Figure 4E**. The experimental information was provided on page 8, line 299-301 in the original manuscript:

[now on page 8, line 327-329 in the revised manuscript] Characterization. Morphological changes in the fibrous substrates were observed by field emission scanning electron microscopy (FE-SEM, Hitachi, S-4800). Samples were precoated with platinum (3 nm) to prevent surface charging during imaging.

5) Figure S9 and discussion in lines 234-235. The authors attribute the loss of fluorescent signal to a non-attachment of dye, but fluorescence quenching by melanin-like materials may also play a role here. The authors should discuss this possibility.

We appreciate the reviewer's comment. We agree with reviewer's suggestion, and think that both the prevention of dye adsorption and fluorescence quenching by melanin-like pigment attenuated the background fluorescence on the pigmented region. We have revised the manuscript as detailed below.

[Before, page 6, line 234-236] Unexpectedly, it was found that the pigment prevented dye adsorption on the A4 paper, minimizing the background during visualizing cells by fluorescent dyes (red: ethidium homodimer-1 for dead cells, green: calcein-AM for live cells) (**Figure S9**).

[After, page 7, line 252-255] Unexpectedly, the pigmented region showed significantly weaker background fluorescence. This may have been due to the prevention of dye adsorption and/or fluorescence quenching by the pigment on the A4 paper (red: ethidium homodimer-1 for dead cells, green: calcein-AM for live cells) (**Figure S15**).

We also revised the legend of **Figure S15**:

[Before] Dye adsorption prevention of PC/DETA-based pigment on A4 paper.

[After] Attenuation of background fluorescence from the pigmented region of A4 paper.

6) I disagree with the argument (lines 266-268) that melanin has rarely been mimicked. Plenty of literature on melanin-like materials exist. Instead, the authors should highlight their own achievement of highly localized synthesis of melanin-like materials.

We agree with the reviewer's comment that there are many reports on melanin-like materials. We meant to communicate the point that most of these studies tried to mimic the ROS-scavenging properties and/or UV-visible-NIR adsorption, and that the "adhesive properties" of melanin in nature have rarely been studied. Our original explanation did not communicate this effectively. We have revised the last paragraph in the discussion with an open question based on this and reviewer 3's comment 9, as detailed below.

[Removed, page 7, line 265-268 in the original manuscript] Mussel adhesive proteins have received great attention as models inspiring the development of polymeric adhesives for biomedical applications. On the other hand, melanin, another example of catechol-containing materials possessing the characteristic underwater adhesive properties in nature, has rarely been mimicked, perhaps due to the uncontrollability of progressive assembly.

[Added, page 7-8, line 292-297] Most studies mimicking melanin have focused on achieving material properties, such as ROS scavenging and UV-visible-NIR adsorption. However, many of the underwater adhesive properties of melanin remain unknown. In melanin, particularly eumelanin, 5,6-dihydroxyindole (DHI) has been reported to play a crucial role in determining its properties. However, our study raises the possibility that DHI may not be important because PC-based precursors yield pigments with both UV-visible-NIR broadband adsorption and underwater adhesive properties. This

requires further investigation.

7) References 27 and 28 (line 82): please check the name of the group.

We apologize for our mistake. We have revised the name of the group (Yu et al. → Weil's group) for references 27 and 28.

Reviewer #3 (Remarks to the Author):

This paper by Jeong et al. is a further contribution by this research group to the exploitation of the adhesive properties of pigments obtained from dopamine or related compounds. In this study in particular a drawback of polydopamine related coatings that is the low site specific adhesion is addressed through a proof of concept approach in which an initiator, actually an oxidizing enzyme, is adsorbed on papers or hydrogels in order to promote pigment formation only on the areas where the initiator/oxidant has been deposited. After a selection of the best performing precursors the method is applied for writing on paper or hydrogels by ink jet distribution of the initiator followed by exposure to the solution of the precursor and hydrogen peroxide. Finally, two applications are reported to exemplify the potential of the so-called PAINT method.

Comment 1: The idea underlying this work may be of interest. As the authors state in the introduction this is not completely new as other related work have appeared though the way the site specificity is achieved is somewhat different. Beside the papers cited please consider also Int. J. Mol. Sci. 2020, 21, 4873 where tyrosinase-loaded alginate spheres, or films deposited on glass or polyethylene allowed PDA pigment formation from tyramine (converted to dopamine by tyrosinase) or also Biofabrication 2020, 12, 035009 for a study of the printability of metal ion crosslinked PEG-catechol based inks.

Thank you for suggesting valuable references that we had overlooked. We have added the two suggested references (now as ref 25 and 28) in the introduction section.

There are several issues that should be considered by the authors

Introduction :

Comment 2: Line 54 and ff.: while the notion that melanin grows on a protein template during melanosome maturation from stage I to IV before their transfer to keratinocytes is well recognized, the idea that this is an example of underwater adhesion is not really supported by solid evidence. Tight adsorption on the protein matrix or even covalent linking of some pigment units to the protein nucleophilic residues are also viable options. Ref 17 and 18 quoted seem not appropriate to melanogenesis in nature as they refer to a model system of amyloid fibers and polydopamine, or cell free melanogenesis in fungi.

We agree with the reviewer's opinion that both physical adsorption and/or covalent linking of the pigment and the protein template may be possible mechanisms. We use the term 'underwater adhesion' to represent all possible types of physical and chemical interaction at the molecular level. We have clarified this in the main text as detailed below.

[Added, page 2, line 59-60] (by physical adsorption and/or covalent interactions at the molecular level)

Regarding references 17 and 18 in the original manuscript, we agree with the reviewer's comment, and have replaced them with the following references:

[revised Ref 17] Fowler, D. M., Koulov, A. V., Balch, W. E., Kelly, J. W. Functional amyloid – from bacteria to humans. *Trends Biochem. Sci.* **32**, 217-224 (2007).

[revised Ref 18] Fowler, D. M., Koulov, A. V., Alory-Jost, C., Marks, M. S., Balch, W. E., Kelly, J. W. Functional Amyloid Formation within Mammalian Tissue. *PLoS Biol.* **4**, e6 (2006).

Comment 3: Line 92-93: Actually the oligomer intermediates arising from oxidation of the precursors are not identified nor characterized in this work so their intrinsic underwater adhesion is just a hypothesis. I recommend to caution this sentence and just report what has been demonstrated.

The HPLC-MS data in **Figure S8** show that oligomeric intermediates formed in the solution-phase reaction between PC and primary amines by HRP/H₂O₂. We suggest that this could be experimental evidence for progressive assembly although they can be just a part of whole intermediates (see the reviewer 3's comment 5).

At this time, we performed an additional experiment to confirm that the UV-vis spectrum of pigment formed on surface through PAINT approach was identical to the pigment synthesized in solution-phase reaction (newly added **Figure S10**). In addition, no diffused pigment was observed in the solution during the PC-based on-surface pigment generation through PAINT method, (see newly added **Figure S7** following reviewer 1's comment 2). Therefore, we believe it is reasonable to say that the detected oligomers were localized on the surface in our PAINT approach.

We have added the text describing **Figure S10** as follows:

[Added, page 5, line 199-201] The comparison of solution-synthesized pigment and the pigment fabricated on a gel by PAINT approach indicates that the detected intermediates were successfully localized on the surface (**Figure S10**).

[Newly added **Figure S10**]

Figure S10. Comparison of solution-synthesized pigment and the pigment fabricated on a gel by PAINT approach.

Results:

Comment 4: It is not clear why the authors that intended to investigate melanin like pigments select in their preliminary screening a number of substrates that can not give rise to melanin type pigments like a series of anilines, resorcinols, hydroxybenzoic acids. The oxidation chemistry of these compounds is well known and most the observations reported could be predicted based on literature data or even chemical reasoning of the structure of these potential precursors. Also, there are some intriguing finding e.g. 6-hydroxydopamine appears red in color before oxidation whereas it should exhibit absorption only in the UV region (it seems it is already oxidized before exposure to the oxidant). It is very difficult to follow the text describing figure 2 as the abbreviations of the compounds tested are not reported under the structures (available only in the SI figures).

We investigated whether the type of pigment is crucial for surface localization (i.e., whether the melanin-like progressive assembly is crucial for surface localization by underwater adhesion, or if it could simply be achieved by the monomeric oxidized products). As shown in **Figure 3B** and **3D**, localization was not only achieved for compounds like dopamine, pycatechol, and pyrogallol (i.e., compounds with a melanin-like with broadband absorption spectrum), but also for compounds like

catechin (which results in water-soluble pigments) and 4-AP (which results in aggregates rather than a nanometer-thick coating, as shown in **Figure 3E**). Although the localization could also be achieved by pigments other than melanin-like ones, the nanoscale behaviors differed between the two categories of compounds, as shown in **Figure 3D** and **3E**.

We also agree with the reviewer's comment that 6-hydroxydopamine seems to have already oxidized before adding HRP. Accordingly, we have revised the related sentence in the text as follows:

[Before, page 4, line 115] intrinsic background signal of the precursor itself or a minimum response after enzymatic conversion

[After, page 4, line 118-119] intrinsic background signal of the precursor itself, autooxidation without enzymatic conversion, or a minimum response to HRP/H₂O₂

We have also added abbreviated names to the structures shown in **Figure 2**.

[Revised Figure 2A]

Comment 5: Line 182 and ff :The role of amine compounds in pigment darkening. The aus state “we isolated a few reactions based on several identified water soluble intermediates (in Fig S7)” the sense of this sentence is quite obscure so as data reported in table of fig S7 . Please show some representative structures in the main text. Also, a darkening that is a bathochromic shift is more likely the result of the addition of primary amino groups to catechols (as is the case of PC) rather than Schiff base formation. Possibly oxidation products of aminocatechols contribute to such darkening process as well. The meaning of sentences lines 188-193 is not clear: why the products resulting from Michael addition should not be detected by HPLC analysis? under the acid elutographic conditions the adducts should be protonated.

As mentioned earlier in response to Reviewer 2's comment 3, different structures may have the same molecular mass. For this reason, we were careful not to conclude the formation of any specific structures based solely on the detected mass. To avoid confusion among readers, we have revised **Figure S8** to include plausible structures and have revised the relevant sentence in the text as follows:

[Before, page 5, line 184-186] Further studies via HPLC–MS analysis were performed to identify the key reactions between PC and these amine additives, and we isolated a few reactions based on several identified water-soluble intermediates (**Figure S7**).

[After, page 5, line 184-186] Several oligomeric intermediates were detected by HPLC-MS analysis of the solution-phase reaction of PC and amine additives with HRP (**Figure S8**).

[Revised, supporting information, Figure S8]

Figure S8. HPLC–MS analysis of water-soluble intermediates identified from the PC-based pigmentation with three different amine additives.

The HPLC-MS data for the compounds formed upon reaction between PC and the primary amines (DETA and EDA, not DEA) is consistent with Schiff-base formation rather than a Michael addition reaction. However, we completely agree with the reviewer’s comment, and we did not intend to state that Schiff-base formation is the main reason for pigment darkening. Our previous explanation did not communicate this effectively.

We re-analyzed the mixture and found that some aggregates filtered out before HPLC-MS analysis (newly added **Figure S9**). We think that Michael-addition products could have been present in these particles, but the intermediates in the particles were hard to isolate because of their melanin-like strong interactions. Therefore, we decided not to use the HPLC-MS data as experimental evidence for the mechanism of reaction between PC and primary amines. Instead, we added references [47-49] that support the occurrence of both Michael addition and Schiff-base formation together.

Nevertheless, the HPLC-MS data are still important because they provide direct experimental support for progressive assembly, showing at least some of the oligomeric intermediates (see the reviewer 3’s comment 3). Therefore, we revised the main text related to **Figure S8** to clarify that the HPLC-MS data provide experimental support for the formation of oligomeric species during progressive assembly, instead of the previous argument that Schiff-base formation is the key to pigment darkening. The revised sentences text is shown below.

[Before, page 5, line 190-193] Since HPLC–MS analysis only detected the water-soluble components, Michael addition could not be completely rejected from the reactions of PC with DETA or EDA. However, it can at least be said that Schiff base formation would be the key to spectral changes in PC-based pigmentation by additives containing primary amine groups.

[After, page 5, line 189-199] In fact, it has been reported in many studies that both Michael addition and Schiff-base formation occur together upon reaction between catechol and amine groups [47-49]. Under our experimental conditions, only Schiff-base products were detected; however, Michael addition cannot be completely rejected during the reactions of PC with DETA or EDA because aggregates were found to form in the solution (**Figure S9**). These aggregates were difficult to break down into individual intermediates and were thoroughly filtered out before HPLC–MS analysis.

Therefore, it is still possible that the Michael-addition products could be formed but not detected because they were contained in the aggregates. In other words, the oligomeric intermediates detected by HPLC-MS only provide experimental support for progressive assembly, and do not serve as evidence for the dominance of Schiff-base formation in the reaction between PC and primary amines.

[Added references]

[47] Yang, J., Saggiomo, V., Velders, A. H., Stuart, M. A. C., Kamperman, M. Reaction Pathways in Catechol/Primary Amine Mixtures: A Window on Crosslinking Chemistry. *PLoS One* **11**, e0166490 (2016).

[48] Alfieri, M. L., Panzella, L., Oscurato, S. L., Salvatore, M., Avolio, R., Errico, M. E., Maddalena, P., Napolitano, A., Ball, V., d'Ischiam M. Hexamethylenediamine-Mediated Polydopamine Film Deposition: Inhibition by Resorcinol as a Strategy for Mapping Quinone Targeting Mechanisms, *Front. Chem.* **7**, 407 (2019).

[49] Qiu, W.-Z., Wu, G.-P., Xu, Z.-K. Robust Coatings via Catechol–Amine Codeposition: Mechanism, Kinetics, and Application. *ACS Appl. Mater. Interfaces* **10**, 5902-5908 (2018).

[Newly added Figure S9]

Figure S9. Aggregates formed during the solution-phase reaction of PC and amine additives with HRP. The aggregates were thoroughly filtered out before HPLC-MS analysis.

Comment 6: Line 203 why the authors expected that BSA should be able to “abstract “in situ-grown oligomers before attachment to HRP”. Any evidence from previous studies?

We are aware of one study that indicated that proteins, such as albumin, in solution accelerate aggregate formation in solution but slow down the growth of films on solid surfaces (Armelle Chassepot, Vincent Ball, *J Colloid Interface Sci*, 2014, 414, 97-102).

For clarity, we have changed the word “abstract” to “suppress the coating and growth”. We also added this reference in the main text with the following revision to the text:

[Before, page 5, line 202-204] Physical blocking of PC/DETA-based pigment attached to HRP was further confirmed by a competitive reaction with BSA, which can abstract in situ-grown oligomers before attachment to HRP (**Figure 4C**). The activity ~

[After, page 6, line 209-211] Physical blocking after the attachment of PC/DETA-based pigment to HRP was further confirmed by adding BSA in the precursor solution, which can suppress the surface growth of pigments [50]. As shown in **Figure 4C**, the activity of ~

[Before, page 17, legend of Figure 4C] BSA that can competitively abstract in situ-grown oligomers before attachment to HRP.

[After, page 19, legend of Figure 4C] BSA that can suppress the attachment of pigment to HRP.

In addition, the diffusion of pigment into the solution was only detected in the presence of BSA, as shown in newly added Figure S11, indicating that BSA abstracts the pigment from the surface. We've made the following changes to the manuscript to present this result:

[Newly added Figure S11]

Figure S11. UV–visible spectrum of pigment that had diffused into the solution during surface-initiated pigment fabrication by the PAINT approach in the presence of BSA in the precursor solution.

[Added, page 6, line 213-214] In this case, the pigment that diffused into solution as a result of the interaction with BSA was detected (Figure S11).

Comment 7: Line 206 and ff figure 4: it is rather intriguing to see the arrow correlating the first panel in fig 4D to the second one with the label encrypted, while it appears that the experiment described is just decrypting the invisible logo after exposure to the catechol solution.

We agree with the reviewer's comment. Figure 4D demonstrates pattern generation by conventional ink-jet printing. Our system may be used for information encryption or decryption, but we don't think we have enough experimental data to support that. Therefore, we have revised the figure as shown below to not mention encryption or decryption, and have revised the manuscript text accordingly. The pigment image was also changed in response to reviewer 1's comment 4.

[Revised legend of Figure 4D] Demonstration of pigmented pattern generation by inkjet-printed HRP.

[Before, page 5-6, line 206-208] Conventional inkjet printers can be employed to prepare encrypted patterns by using HRP as an invisible security ink on two-dimensional membranes, which are readily decrypted as melanin-like pigmented patterns by simply immersing in aqueous solution of PC and DETA.

[After, page 6, line 215-216] Conventional inkjet printers can be used to prepare complex patterns by using HRP as an invisible ink on two-dimensional membranes, which can be readily visualized through PAINT method.

Comment 8: A general consideration that the authors should address and comment is the actual stability of the solutions of the catechols containing hydrogen peroxide which are used for dipping the HRP printed paper/hydrogel

Hydrogen peroxide oxidizes catecholic precursors, and this process is catalyzed by HRP. Therefore, we made a fresh aqueous solution containing catecholic precursors and hydrogen peroxide immediately before dipping the HRP-printed surfaces. We added a description of this process in the experimental section as follows:

[Added, page 8, line 315-317] Because hydrogen peroxide oxidizes the precursor (although the reaction rate is much slower than in the HRP-catalyzed system), hydrogen peroxide was added to the precursor solution immediately before dipping the HRP-loaded surfaces.

Comment 9: Discussion: I would recommend to comment also on the difficulties intrinsic in the PAINT method, to caution the versatility of the proposed methodology. Also the view that melanin is just an example of a catechol material is rather simplistic as in the animal kingdom melanins are indeed more complex as the authors state in the introduction and such complexity is the result of the chemistry of 5,6-dihydroxyindole biosynthetic precursors not any catechol system.

We appreciate the reviewer's comment, which has helped us improve the discussion section. We couldn't identify any intrinsic difficulties in our PAINT approaches, but we wanted to mention that there are many alternatives to HRP used as an initiator, such as nanocatalysts and DNAszymes, which can further extend the range of applications of our PAINT method. We added the following text to communicate this.

[Added, page 7, line 285-289] We selected HRP as our model initiator to demonstrate the PAINT approach. However, catalytic nanomaterials and enzyme-mimicking macromolecules like DNAszymes can also be adapted into PAINT as surface-loaded initiators. These variations in initiators further extend the range of applications of our PAINT method by possessing better stability than HRP for long-term storage and/or increasing the pigmentation speed.

In addition, following this comment and reviewer 2's comment 6, we removed the last two sentences from the last paragraph. Instead, we now pose an open question about the implications of our results on our understanding of natural melanin. 5,6-dihydroxyindole (DHI) has been reported to contribute to the properties of melanin, particularly eumelanin. However, our study based on pyrocatechol with diamine suggests that DHI may not be essential, because both melanin-like broadband adsorption and underwater adhesion were achieved. This requires substantial further investigation. We now mention this in the last paragraph of the discussion section.

[Removed, page 7, line 265-268 in the original manuscript] Mussel adhesive proteins have received great attention as models inspiring the development of polymeric adhesives for biomedical applications. On the other hand, melanin, another example of catechol-containing materials possessing the characteristic underwater adhesive properties in nature, has rarely been mimicked, perhaps due to the uncontrollability of progressive assembly.

[Added, page 7-8, line 292-297] Most studies mimicking melanin have focused on achieving material properties, such as ROS scavenging and UV-visible-NIR adsorption. However, many of the underwater adhesive properties of melanin remain unknown. In melanin, particularly eumelanin, 5,6-dihydroxyindole (DHI) has been reported to play a crucial role in determining its properties. However, our study raises the possibility that DHI may not be important because PC-based precursors yield pigments with both UV-visible-NIR broadband adsorption and underwater adhesive properties. This requires further investigation.

REVIEWERS' COMMENTS

Reviewer #1 (Remarks to the Author):

My concerns have been addressed. I have no further questions.

Reviewer #2 (Remarks to the Author):

The authors have fully addressed my comments. I congratulate them on their excellent work. The manuscript is now suitable for publication.

Reviewer #3 (Remarks to the Author):

The authors have addressed most of the issues raised, and tried to provide additional experimental evidence in support to their conclusions. The picture of melanin role remains quite confusing, e.g. eumelanins are not adhesive per se like mussel byssus nor is 5,6-dihydroxyindole chemistry the key to the underwater adhesive properties of melanins as the authors state “ However, many of the underwater adhesive properties of melanin remain unknown. In melanin, particularly eumelanin, 5,6-dihydroxyindole (DHI) has been reported to play a crucial role in determining its properties. However, our study raises the possibility that DHI may not be important because PC-based precursors yield pigments with both UV–visible–NIR broadband adsorption and underwater adhesive properties” Indeed it was shown that addition of diamines is required to impart adhesive properties to melanins from 5,6-dihydroxyindole precursors (e.g Journal of Colloid and Interface Science 624 (2022) 400–410 in addition to ref 48 of the revised ms) . In this regard the paragraph of the introduction (lines 54-60) and the new refs quoted remain an interpretation by the authors not really supported by experimental evidence.

REVIEWERS' COMMENTS

Reviewer #1 (Remarks to the Author):

My concerns have been addressed. I have no further questions.

→ We thank the reviewer for reading this article carefully and giving instructive comments.

Reviewer #2 (Remarks to the Author):

The authors have fully addressed my comments. I congratulate them on their excellent work. The manuscript is now suitable for publication.

→ We thank the reviewer for the recommendation of publication.

Reviewer #3 (Remarks to the Author):

The authors have addressed most of the issues raised, and tried to provide additional experimental evidence in support to their conclusions. The picture of melanin role remains quite confusing, e.g. eumelanins are not adhesive per se like mussel byssus nor is 5,6-dihydroxyindole chemistry the key to the underwater adhesive properties of melanins as the authors state “However, many of the underwater adhesive properties of melanin remain unknown. In melanin, particularly eumelanin, 5,6-dihydroxyindole (DHI) has been reported to play a crucial role in determining its properties. However, our study raises the possibility that DHI may not be important because PC-based precursors yield pigments with both UV–visible–NIR broadband adsorption and underwater adhesive properties” Indeed it was shown that addition of diamines is required to impart adhesive properties to melanins from 5,6-dihydroxyindole precursors (e.g Journal of Colloid and Interface Science 624 (2022) 400–410 in addition to ref 48 of the revised ms) . In this regard the paragraph of the introduction (lines 54-60) and the new refs quoted remain an interpretation by the authors not really supported by experimental evidence.

→ Based on the reviewer’s comment, we have revised the sentences in the introduction (lines 54-60) to a clearer explanation of how natural melanogenesis inspired the development of our novel fabrication method for polycatecholic materials. We acknowledge the reviewer's point that our previous sentences included our own interpretation of unexplored melanin adhesion. We have now revised the sentences to only mention what are addressed in references with experimental supports for the concept of localized melanin synthesis on amyloid templates, which are believed to prevent the diffusion of reactive intermediates in nature. This is precisely what we have mimicked from natural melanogenesis in our PAINT method. The revised sentences are as follows:

[Before] By revisiting melanogenesis in nature, we were able to draw attention to the previously neglected adhesive nature of melanin. As shown in **Figure 1B**, melanin synthesis

in nature is not a simple solution-phase reaction, which is different from other polymerization reactions. Someone may consider spherical particulates as forms of natural melanins, but in fact, these are mostly melanin-containing melanosomes, not pure melanin pigments. True melanin pigments are grown attached to amyloid fibrils, the template, through a given underwater adhesion (by physical adsorption and/or covalent interactions at the molecular level) and therefore are hard to isolate [17,18].

[After] By revisiting melanogenesis in nature, we were able to draw attention to the amyloid fibrils that act as templates, as shown in **Fig. 1b**. Melanin synthesis in nature is not a simple solution-phase reaction, which is different from other polymerization reactions. Someone may consider spherical particulates as forms of natural melanins, but in fact, these are mostly melanin-containing melanosomes, not pure melanin pigments. Instead, true melanin pigments are found to be grown attached to these amyloid fibrils in melanosomes. More interestingly, the localized synthesis on templates was mediated by the catalytic activity of fibrils that accelerates melanin synthesis and is believed to prevent the diffusion of reactive intermediates out of the melanosome by binding and sequestering [17,18].